# Functional safeguards for conservation: Identifying climate change refugia for frugivorous and nectarivorous birds in a degraded area of Colombia

**Fausto Sáenz-Jiménez**[1]*, **María Alejandra Parrado-Vargas**[2], **José F. González-Maya**[2,3], **Juan Emiro Carvajal-Cogollo**[1]

**1** Museo de Historia Natural Luis Gonzalo Andrade, Facultad de Ciencias, Grupo de investigación Biodiversidad y Conservación, Universidad Pedagógica y Tecnológica de Colombia, Tunja, Boyacá, Colombia, **2** Proyecto de Conservación de Aguas y Tierras – ProCAT Colombia, Bogotá, Colombia, **3** Área en Biología de la Conservación, Departamento de Ciencias Ambientales, División de Ciencias, Biológicas y de la Salud, Universidad Autónoma Metropolitana, Unidad Lerma, Lerma de Villada, C. P, Estado de México, México

\* jfgonzalezmaya@gmail.com

## Abstract

Habitat loss and climate change are major drivers of biodiversity loss, but their synergistic effects and functional perspectives have to be better understood. We employed species distribution models under future contrasting socioeconomic scenarios to assess the impacts of climate change and human footprint on avian frugivore and nectarivore functional groups in the Magdalena Valley, a highly transformed and biodiverse region in Colombia. We constructed the functional groups based on a dissimilarity matrix with 16 anatomical and ecological traits. Two types of future climatic refugia (type 1: areas that will maintain the current climatic conditions and type 2: regions outside the current distribution area that will have the current climatic conditions) were identified to guide conservation efforts for these groups and associated ecosystem services. Of the 27 functional groups identified, 19 are projected to undergo range reductions of 1–75%, with an average upward shift of their climatic niches along the altitudinal gradient of 690 m. Large frugivores from intermediate elevations, such as toucans and cracids, as well as nectarivores with extreme adaptations and specializations, are expected to experience the most severe range reductions. Distributional and altitudinal shifts will lead to spatial reorganization of communities and a reduction or complete loss of functional group richness, particularly in lowland areas. This could impact ecosystem services relevant for degraded area restoration, such as seed dispersal, fruit availability, and pollination of specialized plant species with economic importance. The low representation of future climatic refugia within protected areas highlights the need to incorporate climate change trends into future conservation strategies for these landscapes.

**Data availability statement:** All supporting data necessary to replicate our study is freely available from http://osf.io/j2ykd, and is part of the project "Functional traits and distribution of frugivorous and nectarivorous birds in a degraded area of Colombia" (DOI: 10.17605/OSF.IO/J2YKD). Sáenz-Jiménez, F., Parrado-Vargas, M. A., González-Maya, J. F., & Carvajal-Cogollo, J. E. (2025, May 2). Functional traits and distribution of frugivorous and nectarivorous birds in a degraded area of Colombia. Retrieved from osf.io/j2ykd.

**Funding:** The author(s) received no specific funding for this work.

**Competing interests:** The authors have declared that no competing interests exist.

## Introduction

Habitat transformation driven by land-use change, in synergy with other biodiversity loss drivers, has led to alterations in the structure and composition of biotic communities and has reduced biological diversity [1]. Since 1970, it is estimated that populations of all known mammals, birds, amphibians, reptiles, and fish have decreased by 68% [1]. Additionally, human activities have increased extinction rates by 100–1000 times compared to pre-human levels, leading to a current sixth mass extinction event [2,3].

In addition to habitat loss, climate change can lead to reductions in the geographical ranges of plant, insect, and vertebrate species [4,5] and an increased risk of extinction, potentially putting one in six species at risk, especially in South America, Australia, and New Zealand [6,7]. The modification of geographical ranges and the loss of species will further modulate the structure of communities and ecosystem functioning, with a potential impact on the ability to provide ecosystem services to society [8].

Although habitat loss and climate change have been recognized as two of the main drivers of biodiversity loss [8,9], the synergistic effects of these two drivers on biological populations are still not fully understood [10]. Similarly, it is known that climate stress gradients and human intervention can limit functional diversity [11–13], most studies on the effects of climate change on biodiversity patterns focus on one or more species, and only a few consider their effect from a functional point of view [14–16], and the effect of the contribution of human pressures on functional diversity patterns is also not fully understood today [12,17].

These potential effects on biodiversity highlight the need to develop conservation strategies that explicitly incorporate projections of environmental change into their planning. One approach used to reduce the potential effects of climate change on biodiversity is the protection of climate refugia; localities whose climate in the future will maintain conditions similar to the current ones, allowing the persistence of species over time (*in-situ* refugia), or new places outside the current distribution area that are expected to present those suitable conditions to which the species could naturally disperse (*ex-situ* refugia) [18–20].

Birds are considered good models for evaluating the effects of climate change on biodiversity overall, as they generally follow and respond to climate and environmental changes for establishing their geographical ranges [20–22]. Furthermore, birds are good at reflecting these changes in their ecological interactions and in the structure of their communities [23–25]. Additionally, birds play fundamental roles in ecosystems, including pollination, seed dispersal, and pest control, making them key elements for the maintenance of biodiversity [26]. In particular, frugivorous and nectarivorous birds play a crucial role in ecosystem dynamics, by depending on specific resources such as fruits and nectar, and at the same time, by contributing to the reproduction and distribution of the source plants [27,28]. Considering these arguments, understanding and predicting the effect of climate change and transformation trends on the structure and composition of bird communities and their functional diversity is important for

proposing mitigation and adaptation actions and promoting long-term conservation of species, their ecological functions, and the services they provide for human well-being [18,20].

In this article, we evaluate the potential effects of climate change and habitat transformation, by analyzing functional diversity of frugivorous and nectarivorous birds as a study system in the Colombian Magdalena biogeographic province [29]. This region harbors high levels of biodiversity and endemism and has historically provided significant environmental services for the country [30]. However, the region has suffered from severe transformation, especially considering that it houses approximately 70% of the Colombian population (c.a. 30 million people) [31] and has experienced drastic changes in land use for more than 500 years, with the transformation of more than 40 million ha throughout this basin [32,33]. Using potential distribution models and different climate change scenarios, we explore the variation in the geographical distribution of functional groups of frugivorous and nectarivorous birds under future climate change scenarios. We expected that most functional groups would exhibit varied loss patterns in their geographic distribution, with only a few groups displaying gains, as described for bird species richness at the taxonomic level [34]. Finally, we identify areas of the Magdalena province with potential as future climate refugia for these functional groups and discuss how these changes may impact ecosystem services and what it means for conservation efforts in the face of these changing trends.

## Materials and methods

### Study area

The study was conducted in the Magdalena biogeographic province, located between the western and eastern foothills of the Cordillera Oriental and Cordillera Central of Colombia. Its central axis is the Magdalena River, considered the largest river system in Colombia and one of the longest rivers in the world with a length of approximately 1612 km and a basin of more than 250,000 km$^2$, nearly 21.8% of the Colombian continental surface (S1 Fig) [33,35]. Since pre-Hispanic times, the Magdalena River has been the main source of water and food for the populations settled in its basin. Currently, it is the country's main river trade route, connecting the main urban centers with the Caribbean Coast, and contributes about 80% of its gross domestic product [36,37].

The basin is mainly located in the warm thermal floor, with an average annual temperature ranging between 24 and 28 °C, a bimodal rainfall regime, and precipitation ranging from 3,000–7,000 mm annually. It comprises a wide variety of biomes, including the Magdalena and Caribbean Helobiome, the Andean Upper, Middle, and Lower Orobiomes, the Magdalena and Caribbean Tropical Humid Zonobiome, and the Caribbean Tropical Dry Zonobiome. At the ecosystem level, dense natural high forests stand out for their extension, covering 1,058,191 ha. However, there are also smaller ecosystems present, such as dry forests and Andean and sub-Andean forests. Its geology and hydrological regime have given rise to a series of depressions formed by the basin's largest tributaries, generating an important wetland complex made up of about 205 wetlands [38]. The region is part of the Tumbes-Chocó-Magdalena biodiversity hotspot [39], and heterogeneous environmental conditions have allowed the presence of a great biodiversity with 1,036 species of flora, 368 birds, 129 fish, 30 amphibians, 41 reptiles, and 48 mammals reported for the region [38,40–42].

The growing demand for food and land, driven by population increases since the 1970s, as well as the establishment of hydroelectric power plants and dams, oil and mining extraction, and the expansion of large oil palm plantations, has significantly impacted the region's biodiversity. These activities have contaminated the waters of the river and its main tributaries and have led to land-use changes responsible for the loss of 69% of the Andean forests and 30% of the lowland forests in the basin. Together, these changes contributed to 24% of the total deforestation in Colombia between 2005 and 2010 [33,43,44].

To define the limits of the study area, four ecoregions were used based on the information available on the Ecoregions 2017 platform [45]: i) Magdalena Valley Montane Forests, ii) Magdalena Valley Dry Forests, iii) Magdalena-Urabá Humid Forests, and iv) areas of the Northern Andes páramo ecoregion in the vicinity of the other three ecoregions. These limits

were defined with the aim of having a wide elevational gradient, considering the possibility that species may move to higher altitudes in the face of climate change scenarios [46–48].

## Species data acquisition and filtering

A list of bird species with potential presence for the study area was obtained from national and international repositories (i.e., SIB Colombia, GBIF, VertNet), and from a literature search for the Colombian Magdalena region [40,49–52]. From the 494 general species list with potential presence in the area, frugivorous and nectarivorous species were filtered using the classification provided by the AVONET database [53], resulting in a total of 111 species (i.e., 58 frugivores and 53 nectarivores). We collected recent presence records from the Neotropical region over the last 30 years. This allows us to exclude any records of species that may have been affected by past disturbances in the study area, ensuring a more accurate analysis that reflects current environmental conditions. A quality control process was applied to the dataset, excluding records lacking coordinates, those with uncertain georeferencing, and duplicates within 1 km² grid cells [54,55].

## Functional traits and functional group delineation

Based on the selected species list, we constructed functional trait matrices using the AVONET database [53], which includes average data for various functional traits for most bird species worldwide. The matrix included 16 traits, encompassing anatomical measurements (i.e., bill length-culmen, bill length-nares, bill width, bill depth, tarsus length, wing length, Kipp's distance, length from carpal insertion to tip of first secondary, tail length, biomass) and ecological characteristics (i.e., habitat type, habitat density, migration, trophic level, and lifestyle). These traits are related to different aspects of species ecology, such as movement, feeding, trophic niche structure, and interactions [53].

To construct functional groups, a dissimilarity matrix was created using Gower's distance [56], considering both quantitative and qualitative variables. All variables were rescaled and standardized to avoid the influence of trait scale. The *gawdis* function [57] was used to address the limitation of unequal contribution of different trait types in a multi-trait dissimilarity matrix.

A dissimilarity matrix was created for both frugivores and nectarivores and was used to generate functional groups. This process involved connecting species through clusters within a functional space based on a similarity analysis of species traits [58]. Ward's hierarchical clustering algorithm [59] was employed to define functional groups. Subsequently, the optimal number of groups in the cluster that maximizes inter-group dissimilarity and minimizes intra-group dissimilarity was calculated using the *Nbclust* package [60]. The dendrogram was edited and visualized using the *factoextra* package [61].

Based on the traits of each group, we explored the distribution and variability of the data for each trait to understand the group´s composition and the most relevant traits in this formation using a Principal Component Analysis. Boxplot graphs were created for quantitative variables and scatter plots for qualitative variables to visualize differences between groups.

## Current and future species distribution models

Following the identification of functional groups, we utilized georeferenced occurrence data for each species within a group to model their geographic distribution. These models were built using a suite of bioclimatic variables with a 1 km² resolution obtained from WorldClim. The variables included annual trends and seasonality: Temperature: Bio1 (Mean annual temperature), Bio2 (Mean diurnal range), Bio4 (Temperature seasonality); Precipitation: Bio12 (Annual precipitation), Bio15 (Precipitation seasonality), and the ENSO effect: Bio13 (Precipitation of the wettest month), Bio14 (Precipitation of the driest month) [62]. To further refine the models and account for topographic and habitat suitability, we incorporated the slope extracted from the global Geomorpho90m database [63] and land cover classes at 1 km² resolution, obtained from the EarthEnv project (www.earthenv.org); this project offers continuous land cover information whose suitability for species distribution models is well-established (Table 1) [64]. To avoid correlation issues among the

variables, we conducted a correlation test for each functional group model, excluding any variables that exhibited high correlation coefficients (>0.7, >-0.7) (S2 Fig).

We employed the Maxent algorithm [65,66] within the Wallace platform [67] to build and validate species distribution models for each functional group. We chose Maxent because it effectively models species presence data and generally demonstrates high predictive accuracy [65,66]. The Wallace platform was selected for its ability to facilitate data acquisition and enable users to manage multiple species in one session [67]. To prevent spatial autocorrelation, we applied a minimum distance of 10 km for data thinning. Model calibration involved generating 25 candidate models per group using 10,000 background points, five regularization values (to reduce model complexity), and five feature combinations (linear, quadratic, hinge, product and threshold) for the variable responses [68]. The best model for each species was chosen based on performance metrics like AUC tests, omission reates and AICc delta values [69,70].

For climate projections, two Shared Socioeconomic Pathways (SSPs) were used, which standardize all socioeconomic assumptions (population, gross domestic product, poverty, among others) through modeled representations for each scenario and describe alternative routes for the evolution of future societies in the absence of climate change or climate policies: i) SSP1–2.6, which represents a development based on sustainable practices with low greenhouse gas emissions (2.6 W m-2), and ii) SSP5–8.5, which assumes a fossil fuel-based economy with high greenhouse gas emissions (8.5 W m-2) [71,72]. Climate data for each SSP was downloaded for two time periods (2021–2040 and 2061–2080) at a 1 km² resolution (30 arcseconds; www.worldclim.org).

Two global circulation models were selected, considering their representativeness of the ENSO (El Niño Southern Oscillation) phenomenon [73]: MPI-ESM1.2 HR, which maintains a climate sensitivity that increases with global warming [74,75], and CMCC-ESM2 [76], which shows consistency in its projections of future climate change and represents current climate conditions such as average climate, seasonal patterns, and climate variations [77].

From the resulting models, binary presence/absence maps were constructed using the 10th percentile minimum training presence, which reduces overprediction and is considered appropriate for conservation purposes [78,79]. The potential distribution of the established groups was initially modeled, using the Neotropical region as the calibration area, using the minimum convex hull function with a 1-degree radius for all presence points of the species that made up each functional group. Subsequently, the resulting model was transferred to the study area defined for the Magdalena Valley region, with the aim of considering all possible environments within the area accessible for each species [68].

**Table 1. Variables used to model the geographic distribution of frugivores and nectarivores bird functional groups in the Colombian Magdalena biogeographic province.**

| Type | Variable | Resolution | Functional groups | Source |
|---|---|---|---|---|
| **Climatic** | Bio 1 (Annual mean temperature) Bio 2 (Mean diurnal range) Bio 4 (Temperature seasonality) Bio 12 (Annual Precipitation) Bio 13 (Precipitation of wettest month) Bio 14 (Precipitation of driest month) Bio 15 (precipitation seasonality) | 1 km² | All groups | www.worldclim.org |
| **Topographic** | Slope | 1 km² | All groups | www.portal.opentopography.org |
| **Land-cover** | Evergreen broadleaf trees | 1 km² | Groups of dense forest habitats | www.earthenv.org |
| | Mixed trees | 1 km² | Groups of semi-open habitats | |
| | Shrubs | 1 km² | Groups of semi-open and open habitats | |
| | Cultivated and Managed Vegetation | 1 km² | Groups of human-modified habitats | |

To incorporate the effect of landscape transformation, all generated models were subsequently processed with the 2015 human footprint layer for Colombia [80], a 300 m resolution grid that analyses the change of six variables of land cover transformation between 1970 and 2015. We used two human footprint thresholds for the analysis: high intervention (threshold equal to or greater than 80%) and low to medium intervention (threshold equal to or less than 60%). The models were clipped according to the corresponding human footprint threshold, depending on whether the species were from open or transformed areas, or from dense forest areas. Considering that there were no projections of human footprint for future scenarios, the assumption was used that this human footprint index would remain constant in future scenarios. All analyses were performed in QGis.

### Identification of climatic refugia

The projected models were used to identify two types of climatic refugia within the Magdalena Valley: i) type 1: regions that will maintain the current climatic conditions for the presence of the different functional groups (i.e., *in-situ* climatic refugia) in future scenarios, ii) type 2: regions outside the current distribution area of the groups, which under future scenarios will have the current climatic conditions suitable for the presence of the functional groups (i.e., *ex-situ* climatic refugia) [20]. To identify type 1 regions, the areas retained between the current scenario and future scenarios were extracted; for type 2 regions, the areas with suitable conditions in future scenarios outside the current distribution area were extracted. To define type 1 and type 2 regions, current and future distribution models of each functional group were overlapped, establishing the presence of at least half of the functional groups as the threshold to define the areas of each type [20].

To categorize refugia based on human disturbance levels (low = equal to or less than 60% human foot print index, high = equal to or greater than 80%) and their location within or outside the national protected area system, we intersected the i) previously identified climate refugium, ii) human footprint [80], and iii) protected areas polygons (obtained from the national database; runap.parquesnacionales.gov.co). This information serves as a valuable decision-making tool for prioritizing conservation efforts for species, functional groups, and potential ecosystem services under future climate change scenarios.

## Results

### Functional traits and functional group delineation

Based on the clustering of species using the functional trait matrix, a dendrogram was obtained with 12 groups for frugivorous species and one with 15 groups for nectarivorous species (Fig 1). Detailed ecological descriptions and species composition for each functional group are provided in the supplementary material (S1 Table).

Frugivorous groups exhibited high variability in bill, wing, and tail metrics. These traits were primary drivers of variation, as revealed by principal component analysis (PCA), where the first two components explained 69% of the data (S3 Fig). Nectarivorous groups displayed similar variability in biomass, wing, and tail length, along with secondary feather length and bill depth, which together accounted for 61% of data variation in PCA (S4 Fig).

### Impact of climate change and human footprint on species distribution

Based on the generated models for the current scenario and their comparison with future scenarios, the average range loss across all frugivorous groups is projected to be 9% by 2080 under SSP-126 and 19% under SSP-525. For nectarivorous groups, the anticipated range loss is 11% under SSP-125 and 24% under SSP-525. Out of the 12 frugivorous groups analyzed, 8 groups (comprising 33 species) are expected to experience a reduction in their distribution area of 1–75% due solely to climate change. Similarly, 11 out of the 15 nectarivorous groups (4 species) will face comparable reductions. This trend continues even when considering the additional impact of human activities (Figs 2 and 3).

Among frugivores, Groups 8, 9, and 11 exhibited the most substantial range reductions by 2080 under the SSP5–8.6 scenario as modeled by MPI-ESM1.2 HR and CMCC-ESM2. Specifically, Group 8 (*Euphonia concinna* and *Stilpnia vitriolina*) experienced a 30–54% decline, Group 9 (*Ortalis columbiana*, *O. garrula* and *O. guttata*) contracted by 46–65%, and Group 11 (*Ramphastos ambiguus, R. sulfuratus, R. tucanus* and *R. vitellinus*) decreased by 12–55%. Nectarivorous groups exhibiting the most substantial range reductions included: Group 3 (*Anthracothorax nigricollis, Chrysuronia goudoti, Saucerottia cyanifrons*, and *S. saucerottei*, with a 50–51% decline), Group 11 (*Ensifera ensifera*, experiencing a 55–76% reduction), Group 12 (*Eutoxeres aquila* and *Heliodoxa imperatrix*, with a 47–53% decline), and Group 13 (*Heliangelus exortis*, experiencing a 60–71% reduction) (Figs 2 and 3, S2 Table).



**Fig 1. Dendrogram of clusters depicting** (A) **nectivorous and** (B) **frugivorous bird functional groups in the Colombian Magdalena Biogeographic Province.**

Conversely, several groups exhibited range expansions under future scenarios. Among frugivores, Groups 2, 4, 5, and 7 demonstrated increases. Group 2 comprised intermediate-sized species inhabiting open and semi-open areas (i.e., *A. amazonica, P. cayannensis* and *P. subvinacea*), Group 4 included 14 small to medium-sized forest dwellers, Group 5 that only contained *Crax alberti* (a large forest dweller), and Group 7 that encompassed seven small species from semi-open habitats. For nectarivores, Groups 4, 7, 8, and 9 expanded their ranges. These groups included small hummingbird species from various habitat types. While most groups experienced modest range expansions (less than 16%), Frugivore Group 5 and Nectarivore Group 7 showed substantial increases of up to 75% and 196%, respectively (Figs 2 and 3, S2 Table).

**Fig 2. Area shifts for functional groups of frugivorous birds in different future climate scenarios in the Colombian Magdalena Biogeographic Province.** The panel in which the effect of the human footprint is included is denoted by the initials HFP and grey bars.



All functional groups, except for frugivore Group 5, exhibited upward elevational shifts by an average of 690 meters under the SSP5–8.5 scenario by 2080. Frugivore Group 9 and Nectarivore Group 3 experienced the most substantial elevational shifts, increasing by 1247 m and 1424 m, respectively (Figs. 4 and 5, S3 Table). Conversely, groups restricted to lowland habitats showed minimal elevational changes (Fig 4 frugivores 2 and 5, Fig 5 nectarivores 7 and 8).

Projected shifts in species distributions and altitudinal ranges will reshape avian functional communities as they track changing climatic niches. Concurrently, habitat modification, driven by the combined pressures of climate change and human footprint, will exacerbate these changes. By 2080, widespread declines in frugivore and nectarivore functional group richness are anticipated, with the potential for complete loss in certain areas (Figs 6 and 7). Lowland regions, particularly those adjacent to the Caribbean, are projected to experience the most severe declines, with nectarivores facing a disproportionate impact.

**Fig 3. Area shifts for functional groups of nectarivorous birds in different future climate scenarios in the Colombian Magdalena Biogeographic Province.** The panel in which the effect of the human footprint is included is denoted by the initials HFP and grey bars.



Fig 4. Shifts in the elevation range for functional groups of frugivores birds under the current and 2080 SSP5 8.5 future climate scenarios in the Colombian Magdalena Biogeographic Province.



**Fig 5. Shifts in the elevation range for functional groups of nectarivores birds under the current and 2080 SSP5 8.5 future climate scenarios in the Colombian Magdalena Biogeographic Province.**





**Fig 6. Shifts in frugivorous functional groups distribution caused by climate change and human footprint in the Magdalena Biogeographic Province.** F.G. richness = Functional groups richness.

Fig 7. Shifts in nectarivorous functional groups distribution caused by climate change and human footprint in the Magdalena Biogeographic Province. F.G. richness = Functional groups richness.

**Identification of climatic refugia**

Extensive potential climate refugia were identified for frugivorous birds based on human footprint and protected area status: i) Type 1 refugia: 53,269 km² with high human impact, 13,720 km² with low impact within protected areas, and 45,964 km² with low impact outside protected areas, and ii) Type 2 refugia: 9,590 km² with high human impact, 5,446 km² with low impact within protected areas, and 9,352 km² with low impact outside protected areas (Fig 8). These findings provide a baseline for understanding the spatial distribution of potential refugia and inform conservation prioritization.

Nectarivorous bird groups exhibited similar patterns. Type 1 refugia for nectarivores encompassed 23,247 km² with high human impact, 13,720 km² with low impact within protected areas, and 30,405 km² with low impact outside protected areas. Type 2 refugia totaled 2,410 km² with high human impact, 2,158 km² with low impact within protected areas, and 1,915 km² with low impact outside protected areas (Fig 9).

## Discussion

### Impact of climate change and human footprint on bird functional groups distribution

This study assessed the combined impacts of climate change and habitat transformation on the spatial distribution of frugivorous and nectarivorous bird functional groups within the Magdalena biogeographic province. Recognizing the critical role of these guilds in ecosystem services, such as pollination and seed dispersal [23,81,82], our research departed from previous studies by explicitly considering functional diversity, rather than focusing solely on species-level responses to climate change [14]. This approach offers a novel perspective for integrating functional relationships and ecosystem services into conservation planning.

Our findings indicate a substantial decline in the distribution ranges of many functional groups under future climate change scenarios. Specifically, 19 of the 27 evaluated groups (comprising 77 species) are projected to experience range reductions by 2080 under a fossil fuel-intensive pathway. This aligns with projections suggesting over 50% of species may lose suitable climatic conditions by 2100 under high emission scenarios [4]. Beyond range contractions, functional groups will undergo altitudinal shifts, reshaping community composition and structure. These changes can lead to local extinctions, range shifts, and altered species interactions, ultimately impacting ecosystem functions and services [15,83,84].

Habitat loss and fragmentation, driven by deforestation, further exacerbate these climate-driven impacts. Reduced habitat availability constrains species movement and establishment, while also altering community composition through habitat filtering [85–87]. Our results emphasize the synergistic effects of climate change and deforestation on biodiversity [88], highlighting the need for integrated conservation strategies. Projected changes in functional group distributions indicate potential declines in species richness, particularly in lowland areas and coastal regions of the Magdalena basin. This loss of functional diversity is likely to compromise critical ecosystem services, as suggested by previous studies [89,90].

Frugivore groups experiencing the most substantial range reductions include large-bodied taxa crucial for seed dispersal. Groups 1, 9, and 11, encompassing toucans (*Ramphastos, Andigena, Aulacorhynchus*, and *Pteroglossus*) and cracids (*Aburria, Chamaepetes*, and *Penelope*), are particularly vulnerable. These species disperse seeds of numerous plant families, including pioneer species like *Schefflera morototoni* and keystone genera such as *Ficus* [91–94]. Toucans, in particular, exhibit specialized adaptations for seed dispersal, consuming entire fruits and facilitating long-distance seed dispersal [95–97].

Group 9, comprising chachalaca species (i.e., *Ortalis* spp.), also faces significant range contractions. Like other cracids, chachalacas play a vital role in forest regeneration through seed dispersal, contributing to the dynamics of degraded forests [98,99]. Their diet includes a diverse array of plant species, including both wild and cultivated taxa, further highlighting their ecological importance [99,100]. The Colombian guacharaca (*O. columbiana*), for instance, promotes the dynamics of degraded forests by defecating a large number of seeds of different sizes, moving between primary and secondary forests, and promoting successional processes of regenerating forests [99]. The decline of these frugivore groups



**Fig 8. Future climatic refugia for frugivorous functional bird groups in the Colombian Magdalena Biogeographic Province.** Categories for type 2 refugia are not shown due to the visualization scale. The upper right panel shows the areas of each category in square km. T1 = Type 1 refugia, T2 = Type 2 refugia, H HFP = High Human footprint, L HFP = Low Human Footprint, UPAs = Unprotected Areas, PA = Protected Areas.





**Fig 9. Future climatic refugia for nectarivorous functional bird groups in the Colombian Magdalena Biogeographic Province.** Categories for type 2 refugia are not shown due to the visualization scale. The upper right panel shows the areas of each category in square km. T1 = Type 1 refugia, T2 = Type 2 refugia, H HFP = High Human footprint, L HFP = Low Human Footprint, UPAs = Unprotected Areas, PA = Protected Areas.

will likely compromise ecosystem services such as seed dispersal, forest regeneration, and wildlife food availability, with potential implications for human livelihoods [99,100].

Cracids, a primary food source for many Latin American communities, are facing significant threats from hunting and habitat loss [101,102]. As crucial seed dispersers, particularly for large-seeded species, their decline will impact forest regeneration and ecosystem health. Given their role in supporting both human livelihoods and ecological processes, the loss of these species represents a substantial challenge for conservation. The frugivore guild as a whole plays a vital role in forest recovery, making the projected declines particularly concerning for a highly degraded region like the Magdalena Valley [103]. Restoring these ecosystems will require strategic reintroduction of seed-dispersing species.

Nectarivorous Group 11, represented solely by the long-billed swordbill (*E. ensifera*), exemplifies extreme specialization. This hummingbird's coevolutionary relationship with approximately 37 Passifloraceae species highlights its critical role as an exclusive pollinator [104,105]. The potential loss of *E. ensifera*, for instance, could lead to the extinction of *Passiflora mixta*, underscoring the importance of this hummingbird-plant interaction [106]; the genus *Passiflora* holds economic value in food, medicine, cosmetics, and ornamentals [107]. *E. ensifera* also pollinates other long-tubed flowers, including *Brugmansia*, *Aetanthus*, and *Salvia* species [108]. Similarly, Nectarivore Group 12, including the White-tipped Sicklebill (*E. aquila*), exhibits a strong mutualism with *Centropogon* plants [109], and also pollinates *Heliconia* species [110]. Hummingbirds as a group are highly specialized nectarivores, with diverse beak lengths facilitating resource partitioning [111]. Their role as key pollinators for Neotropical plants is well-established [112] and declines in hummingbird populations and associated habitat loss threaten the integrity of these intricate plant-pollinator networks.

While some authors have suggested that lowland tropical bird species will be the most affected by the contraction of their distribution ranges because of climate change [113], our results draw attention to the cases of groups 5 and 7 of frugivores and nectarivores, respectively. These groups are composed of the Blue-billed Curassow (*Crax alberti*) and the Violet-bellied Hummingbird (*Chlorestes julie*), for which the models under climate change scenarios suggest a significant increase in their distribution area (SI4). Both species exhibit restricted distributions to low elevations (<1000 m.a.s.l.) within the Magdalena Basin (Figs 4 and 5). This lowland specialization likely contributed to their projected range expansion, particularly towards areas like the Mompox Depression. This Caribbean wetland complex, characterized by periodic flooding generated by an inland delta formed by the confluence of four of Colombia's main rivers (i.e., Magdalena, Cauca, San Jorge, and Cesar) [114], may become increasingly suitable under future climate scenarios. Floodplains have been highlighted as important areas for adaptation to climate change due to their mesic and cooler microclimatic conditions compared to adjacent areas, as well as their greater access to and availability of water [115].

This trend contrasts with groups with higher altitudinal ranges, which will experience a greater contraction in their distribution area. This is consistent with previous studies that suggest tropical mountain cloud forests are particularly vulnerable to warming due to the potential reduction of cloud cover, reduced water capture, and drier conditions, which will affect the typical communities of these ecosystems as current climates ascend the altitudinal gradient [113,116].

## Climatic refugia and conservation issues

Our results highlight the importance of addressing climate change and deforestation in an integrated manner in the conservation strategies of frugivorous and nectarivorous birds in areas of conservation relevance, such as the Colombian Magdalena Valley. The interaction between future environmental conditions such as climate and deforestation has already been suggested as a relevant approach for developing area-based conservation strategies, and predicting how and where species will change their distribution ranges emerges as an important challenge for conservation biology [84,113].

The design and establishment of new areas within protected area networks must consider the effects of climate change, as current protected areas generally do not account for potential species distribution shifts under future scenarios [113]. Our results show that only 17% of the type 1 climate refugia identified for frugivores and 31% of such refugia

for nectarivores are included within the current protected area system of the Magdalena Valley. Therefore, several of the functional groups and species evaluated would be poorly protected from the effects of climate change.

The identification of areas of conservation importance has generally focused on taxonomic diversity without considering the interactions and ecological processes that occur between species [117,118]. Understanding the effects of climate change and human footprint on functional groups, as well as predicting which areas will be most affected or where the climatic niches of these species will move, will allow for the conservation not only of the species that make up these groups but also of the ecological processes that occur between these species [119]. This information is a useful tool for guiding decisions and actions that enable the long-term maintenance of ecosystem services associated with frugivorous and nectarivorous birds in the Magdalena province.

This information additionally suggests that the conservation of functional groups of frugivorous and nectarivorous birds, as well as birds in general, cannot rely solely on the establishment of protected areas. Instead, effective conservation strategies must incorporate other alternatives such as improving landscape connectivity and implementing actions that make human-dominated landscapes more habitable for birds, with active involvement of local communities in decision-making [113]. Identifying type 1 and type 2 refugia provides a foundation for informed conservation decision-making. These areas can guide the establishment and expansion of protected areas, such as the Serranía de San Lucas, or inform restoration efforts in degraded refugium like the Mompox Depression (Figs 8 and 9).

The predictions of future climate pattern changes still have high uncertainty, which is exacerbated when these predictions are used in ecological models such as potential distribution models, especially when projected to scenarios on a very broad temporal scale [120,121]. To address these inherent uncertainties in long-term climate projections, this study focused on medium-term scenarios (2040–2080), where confidence is generally higher [120]. Additionally, we employed multiple global circulation models and socioeconomic pathways to assess the robustness of projected distribution changes for functional groups. Future research should incorporate species dispersal capabilities into distribution models to more accurately predict access to potential climate refugia under global change scenarios [84].

Our findings regarding the projected declines in frugivore and nectarivore functional diversity resonate with trends observed in other Neotropical regions. For instance, a study in Brazil [122] has also documented significant range contractions for frugivorous birds due to habitat loss and climate change, particularly affecting large-bodied species crucial for seed dispersal. However, the magnitude of these declines varies across regions, potentially reflecting differences in landscape context, the intensity of anthropogenic pressures, and species-specific vulnerabilities. While our study highlights the vulnerability of lowland specialists in the Magdalena Valley, other research [123] has shown that montane species can also be highly susceptible to climate change due to factors such as cloud forest loss and restricted elevational ranges. These regional differences underscore the importance of context-specific conservation strategies, even within the broader Neotropical biome. Furthermore, the synergistic effects of habitat loss and climate change, as observed in our study, are a recurring theme in the literature [124,125], emphasizing the need for integrated approaches that address both drivers of biodiversity loss.

The projected impacts on key functional groups, such as large frugivores and specialized nectarivores, also have parallels in other tropical ecosystems. A previous study [126] demonstrated the critical role of frugivorous birds in maintaining forest regeneration and carbon sequestration, highlighting the potential consequences of their decline for ecosystem functioning. Similar to our findings, research on hummingbird-plant interactions in the Andes [127,128] has revealed the vulnerability of specialized pollination networks to climate change, with cascading effects on plant reproduction and community dynamics. While the specific plant species and ecological interactions may differ across regions, the overarching message is consistent: the loss of functional diversity can disrupt essential ecosystem services and compromise ecosystem resilience. Therefore, the conservation strategies proposed for the Magdalena Valley, such as the establishment of climate refugia and the improvement of landscape connectivity, have broader relevance for other tropical regions grappling with similar challenges. Sharing best practices and lessons learned across different biogeographic contexts is crucial for maximizing the effectiveness of conservation efforts in the face of global change.



While this study represents a partial exploration of biodiversity's functional dimension of a specific group, it underscores the imperative for comprehensive conservation strategies in the face of climate change. Accumulating evidence underscores the necessity of incorporating taxonomic, functional, and ideally evolutionary perspectives into conservation planning to foster resilience and adaptation to rapidly evolving climate conditions.

## Supporting information

**S1 Figure. Location of the study area (Magdalena province) in Colombia and South America.**
(PDF)

**S2 Figure. Correlation matrix plots for bird functional groups in the Colombian Magdalena biogeographic Province.**
(PDF)

**S3 Figure. Exploration of traits for frugivorous functional groups of birds in the Magdalena province in Colombia.** Boxplot of trait variability for functional groups of frugivorous birds. PCA ordination of trait variability for functional groups of frugivorous birds.
(PDF)

**S4 Figure. Exploration of traits for the nectarivorous functional groups of birds in the Magdalena province in Colombia.** Boxplot of trait variability for functional groups of nectarivorous birds. PCA ordination of trait variability for functional groups of nectarivorous birds.
(PDF)

**S1 Table. Description of frugivorous and nectarivorous functional groups composition and description of traits of birds in the Magdalena province in Colombia.**
(PDF)

**S2 Table. Shifts in the distribution area for all groups of birds in the Magdalena province in Colombia.**
(PDF)

**S3 Table. Shifts in average elevation of birds in the Magdalena province in Colombia.** Values of the functional groups with the greatest variation are underlined.
(PDF)

## Acknowledgments

We are grateful to the Ministry of Science, Technology, and Innovation and the Universidad Pedagógica y Tecnológica de Colombia for their logistical and institutional support in the development of this research.

## Author contributions

**Conceptualization:** Fausto Sáenz-Jiménez, María Alejandra Parrado-Vargas, Juan Emiro Carvajal-Cogollo.

**Data curation:** Fausto Sáenz-Jiménez, Juan Emiro Carvajal-Cogollo.

**Formal analysis:** Fausto Sáenz-Jiménez, María Alejandra Parrado-Vargas, José F. González-Maya, Juan Emiro Carvajal-Cogollo.

**Funding acquisition:** Fausto Sáenz-Jiménez, Juan Emiro Carvajal-Cogollo.

**Investigation:** Fausto Sáenz-Jiménez, María Alejandra Parrado-Vargas, Juan Emiro Carvajal-Cogollo.



**Methodology:** Fausto Sáenz-Jiménez, María Alejandra Parrado-Vargas, José F. González-Maya.

**Project administration:** Fausto Sáenz-Jiménez.

**Resources:** Fausto Sáenz-Jiménez.

**Supervision:** Fausto Sáenz-Jiménez, Juan Emiro Carvajal-Cogollo.

**Validation:** Fausto Sáenz-Jiménez, Juan Emiro Carvajal-Cogollo.

**Visualization:** Fausto Sáenz-Jiménez.

**Writing – original draft:** Fausto Sáenz-Jiménez, María Alejandra Parrado-Vargas, José F. González-Maya, Juan Emiro Carvajal-Cogollo.

**Writing – review & editing:** Fausto Sáenz-Jiménez, María Alejandra Parrado-Vargas, José F. González-Maya, Juan Emiro Carvajal-Cogollo.

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
