## [Decision Letter · Decision Letter 0]

21 Aug 2024

PONE-D-24-31667Functional safeguards for conservation: identifying climate change refugia for frugivorous and nectarivorous birds in a degraded area of ColombiaPLOS ONE

Dear Dr. González-Maya,

Thank you for submitting your manuscript to PLOS ONE. After careful consideration, we feel that it has merit but does not fully meet PLOS ONE’s publication criteria as it currently stands. Therefore, we invite you to submit a revised version of the manuscript that addresses the points raised during the review process. Your submission has been reviewed by two external reviewers. I also made my review.

Reviewer 1 suggested Major Revision and provided some compliments.

His/her major concerns are relative to the need of citation of more studies in the Introduction and Discussion, and the need of more clarity in the Methods. Also, there are numerous corrections and suggestions regarding all parts of the manuscript.

Reviewer 2 suggested Minor Revision and provided some compliments.

He/she is mainly concerned with the need of more clarity in the Methods, and some improvement in the Discussion. Also, there is a large set of minor suggestions regarding mainly the use of specific terms and writting.

My own review also provided some suggestions and corrections. Please find it below.

With this, I believe that, after corrections, this manuscript can be published in PLOS ONE.

Thus, I suggest that it keeps in the evaluation process for publication, after Major Revision.

Please let me know if you have any doubts regarding this submission.

Please submit your revised manuscript by Oct 05 2024 11:59PM. If you will need more time than this to complete your revisions, please reply to this message or contact the journal office at plosone@plos.org . Please include the following items when submitting your revised manuscript:

We look forward to receiving your revised manuscript.

Kind regards,

Dárius Pukenis Tubelis, Ph.D.

Academic Editor

PLOS ONE

Journal Requirements:

2. Please note that your Data Availability Statement is currently missing the repository name, the DOI/accession number of each dataset or a direct link to access each database. If your manuscript is accepted for publication, you will be asked to provide these details on a very short timeline. We therefore suggest that you provide this information now, though we will not hold up the peer review process if you are unable.

3. In the online submission form, you indicated that your data will be submitted to a repository upon acceptance.  We strongly recommend all authors deposit their data before acceptance, as the process can be lengthy and hold up publication timelines. Please note that, though access restrictions are acceptable now, your entire minimal  dataset will need to be made freely accessible if your manuscript is accepted for publication. This policy applies to all data except where public deposition would breach compliance with the protocol approved by your research ethics board. If you are unable to adhere to our open data policy, please kindly revise your statement to explain your reasoning and we will seek the editor's input on an exemption. 

4. We note that Figures 5 to 8 in your submission contain map images which may be copyrighted. All PLOS content is published under the Creative Commons Attribution License (CC BY 4.0), which means that the manuscript, images, and Supporting Information files will be freely available online, and any third party is permitted to access, download, copy, distribute, and use these materials in any way, even commercially, with proper attribution. For these reasons, we cannot publish previously copyrighted maps or satellite images created using proprietary data, such as Google software (Google Maps, Street View, and Earth). For more information, see our copyright guidelines: http://journals.plos.org/plosone/s/licenses-and-copyright.

1) You may seek permission from the original copyright holder of Figures 5 to 8 to publish the content specifically under the CC BY 4.0 license.  

2) If you are unable to obtain permission from the original copyright holder to publish these figures under the CC BY 4.0 license or if the copyright holder’s requirements are incompatible with the CC BY 4.0 license, please either i) remove the figure or ii) supply a replacement figure that complies with the CC BY 4.0 license. Please check copyright information on all replacement figures and update the figure caption with source information. If applicable, please specify in the figure caption text when a figure is similar but not identical to the original image and is therefore for illustrative purposes only.

**Additional Editor Comments:**

Review by the editor (Dárius Tubelis):

Title page. Please check again the instructions regarding the authorship and affiliations.

Affiliation: start with the Department, and then the University, to the country (see the wrong item 1).

Abstract. Line 22. Is "poorly" really correct here ? Maybe a too strong term.

About line 25 and ahead. Could you provide 1-2 sentences regarding the methods of your study ?

Introduction

You can indent the first line of the paragraphs, except the first one. (also check this for other sections of the manuscript).

The Introduction is of good lenght and quality but the number of cited studies is considerably low.

You tended to cite only one study per sentence, but you can easily

increase it to 2-3 citations per brackets by making a new search in Scopus.

This would increase the probability of citing the main relevant papers on this topic. Consider the PLOS ONE international audience.

With this improvement, the numbers of different studies would change here and in the References Section. Please check carefully later, if you change them.

You can make this improvement easily, by making little or no changes in your current text.

Line 85. Add "Colombia".

Methods

Study Area

Well done; well written.

Species Data

Line 142. You can add a reference that you used to classify these species as frugivores and nectarivores.

Line 150-154. But are the aspects of these trais in the objectives ?

Distribution Models

Line 185. Add the valley/province and Colombia to the Table title.

Table 1. Column "Variable". Should "tres" be "trees" ?

Results

Lines 262-263. "reduction of between" sounds worng. Maybe you can write "reduction of "1-75%".

Line 269. Fix the "and" in italics. The same for line 278.

Figures 2, 3 and 4. Their captions. Add the valley/province and Colombia in the end of the first sentence. Check this for all figures and tables.

Line 314. You have to bring the captions of figures 5 and 6 here, after this paragraph where you firstly cited them.

Discussion

Within brackets, you are adding a space between consecutive numbers. It should not occur. Please check throughout the manuscript. It should be, e.g., [2,3].

You have to divide the Discussion in sub-sections with subtitles. Preferably, if you follow those of the Results, and one more regarding Conservation Issues.

Keep your mind always with an international readership.

References

Journal titles should be abbreviated.

When citing the pages of articles, you have to use a long dash, not hyphen between numbers.

Please check again if everything is in agreement with the Instructions before sending the corrected version.

Supporting Information

Please make sure that the captions and titles inform that you studied birds, in the Magdalena valley/province, in Colombia.

Figures

It is difficult to read what is written there. Maybe only a problem due download. It will be checked during the next stages.

Figures 3 and 4. I did not understand why you used letters here, instead of the numbers of each group.

Figures 5 and 6. These two black circles are useless as they are. Eliminate them or add the names of cities. Explain in the captions the meaning of "F. G. richness".

Figures 7 and 8. Add Venezuela and move Colombia to the right (gray area). Put them in bold and bigger than the names of the two cities.

Figures 7 and 8. I did not understand why you have two color legends in each figure.

Dárius

Reviewers' comments:

Reviewer's Responses to Questions

**Comments to the Author**

1. Is the manuscript technically sound, and do the data support the conclusions?

Reviewer #1: No

Reviewer #2: Yes

2. Has the statistical analysis been performed appropriately and rigorously? 

Reviewer #1: No

Reviewer #2: Yes

3. Have the authors made all data underlying the findings in their manuscript fully available?

Reviewer #1: Yes

Reviewer #2: Yes

4. Is the manuscript presented in an intelligible fashion and written in standard English?

Reviewer #1: Yes

Reviewer #2: Yes

5. Review Comments to the Author

Reviewer #1: Dear Editors,

Thank you for the opportunity to review the manuscript titled " Functional safeguards for conservation: identifying climate change refugia for frugivorous and nectarivorous birds in a degraded area of Colombia" submitted to Plos One.

In this manuscript, the authors group frugivorous and nectarivorous bird species occurring in a province in Colombia into functional subgroups to evaluate the impacts of climate change on species distributions. While assessing the effects of climate change on functional groups of birds is relevant for biodiversity conservation in biodiversity hotspots, the study lacks novelty. Numerous articles have already been published on the impacts of climate change on bird distributions. Although the authors claim there are very few studies exploring the effects of climate change on functional groups of birds, this assertion is questionable; I found several articles on this topic with a quick Google search (see specific comments below).

The authors mention that they also evaluated the impacts of deforestation on species occurrences, but their methodology lacks clarity. They state they used a footprint layer for this analysis, yet they do not explain essential characteristics of this layer (e.g., whether it allows for species occurrence in non-forested areas, its resolution, etc.).

Similarly, the authors omit crucial information regarding the methods they used to project species' potential distributions. I also believe that their models could be significantly improved (please see specific comments below).

The discussion is weak. Although it is natural for the study to focus on the implications of the results within the study area, the authors have done so excessively. Given that the manuscript was submitted to an international journal, it should include more comparisons with studies conducted in other tropical regions. In this regard, I have suggested some relevant studies in the comments below.

The authors also fail to discuss the implications of their results for the potential movement of species under future scenarios. For instance, is the establishment of ecological corridors relevant for species movement within the landscape? Where could these potential corridors be implemented?

Specific Comments:

Title: I suggest removing "in a degraded area of Colombia." Including regional characteristics in the title might not appeal to an international audience, although this information should be included in the abstract.

Figures: The text in the figures, especially in Figures 1, 2, 3, and 4, is difficult to read. This might be due to the low-resolution images required by the submission system, but I couldn't properly review them. Additionally, is there a reason why the figure legends are spread throughout the manuscript? This arrangement makes them hard to find. Please consider adding a small inset map of Colombia showing the location of the study area. For readers unfamiliar with Colombia's geography (like myself), the location of Magdalena Province might not be immediately clear.

Lines 34-35: I suggest excluding "for other wildlife."

Line 35: Consider removing "highly," as it sounds a bit vague.

Line 35: Replace "value" with "importance."

Keywords: Unless it is a journal requirement, I suggest using lowercase for all keywords and omitting the full stop.

Lines 59-60: "and rarely consider their effect from a functional point of view"—but see the following studies:

https://doi.org/10.1016/j.pecon.2023.12.002

https://doi.org/10.1111/ele.13830

https://doi.org/10.1111/gcb.16723

https://doi.org/10.1038/s41598-019-53409-6

Lines 114-117: Instead of providing biome names, it might be more informative to describe the vegetation types (e.g., forests, savannas, shrublands, etc.).

Lines 140-142: From where did you obtain the species list of 494 species that may occur in the region? Which classification system did you follow to categorize them as frugivorous or nectarivorous? Please clarify, even if it was based on personal knowledge.

Line 143: Why did you choose a 30-year timeframe?

Lines 145-146: Does this mean you retained species records that are more than 1 km apart? If so, could this result in occurrence points being, for example, 1.5 km apart or slightly more? This could lead to high spatial correlation of your occurrence points, potentially causing model overfitting and biasing the results toward regions with more species occurrence data.

Lines 150-154: Please clarify why you selected those particular measures as proxies for functional traits.

Lines 176-178: Please explain why you chose these bioclimatic variables. How did you handle correlation among variables?

Line 181: Is your land cover class variable categorical? It's generally not advisable to use categorical variables in species distribution modeling (SDM) studies.

Line 187: Why did you choose Maxent? Why not include other algorithms and perform an ensemble approach?

Lines 189-190: "five regularization values, and five feature combinations"—I don't understand what this means; please clarify.

Line 191: I'm not sure if AUC is the best evaluation metric. Why didn't you use TSS? https://doi.org/10.1016/j.ecolind.2022.109830

Lines 201-206: How did you perform the ensemble of your global circulation models?

Lines 211-212: Was the convex hull created for each species individually? If so, I agree. Please clarify.

Lines 215-219: I didn't understand how you considered the "effect of landscape transformation." Please provide more information about the variable you used and your analysis.

Lines 245-248: The number of species in each group, as I summed from Table S1 and Figure 1, was 58 for frugivores and 53 for nectarivores. In line 140, you mention that you estimated the distribution of 494 species. Please clarify.

Line 264: I suggest moving Figure 2 to the Supplementary Material.

Lines 267-275: Since species in each group can be seen in Figure 1 and Table S1, I suggest removing species names from the text.

Reviewer #2: This is a robust analysis in an important biodiversity hotspot that deserves more attention. The SDM methodology is traditional but sound. The focus on functional groups is a nice novel addition to this deep area of research/analysis.

I have some medium-sized comments and then a lot of minor wording suggestions below.

1. In general the manuscript is really well written but there are some sections that could be clearer. This could be accomplished by doing a close read and trying to get rid of extra words. I've noted some below. Here's another example: Line 316: "Analysis of projected climate change scenarios identified potential refugia for frugivorous birds. We categorized these refuges based on human footprint and protected area status" could be more directly reworded as "Extensive potential climate refugia were identified for frugivorous birds: Type 1 refugia:...

2. The commonly used terms in the (English) literature now are climate refugia and climate change

refugia. Best to pick one and use throughout.

3. The Methods are not detailed enough and too conceptual. I gave some examples below.

4. Discussion could use some more caveats. For example, related to Line 221: how does the 'assumption was used that this human footprint index would remain constant in future scenarios.' affect the results? Also (~272-274), need to note that SSP585 is increasingly considered unlikely (this is new science) so these results are low probability, worst case scenario. On that point, the SSP 126 results should be showcased a bit more in the text to show the range of possibilities. Also, Lines 321-322 in the Results belong in the Discussion.

Minor suggestions:

Line 21: Suggest 'drivers of the biodiversity crisis' to avoid saying loss again

Line 29: Use altitude or elevation consistently throughout (apparently elevation is the more appropriate term)

Keywords: I suggest simply 'functional group' since 'Avian functional groups' is likely too specific to be a useful keyword term

Line 44: Insert comma before since

Line 45: reword - can delete 'the population sizes of' or reword as 'there has been a 68% reduction in the populations of all (known) mammals..." nearly is not needed unless you want to say 70%

Line 48: wording makes it clear that the previous paragraph was supposed to be about habitat loss, but besides for the opening sentence, the paragraph just seems to be about the biodiversity crisis in general.

Line 57: change comma after however to semi-colon

Line 58: delete 'relative approaches to'

Line 64: change mitigate to reduce - avoid using the term mitigate in the climate change context when not referring to greenhouse gases to avoid confusion

Line 71: replace semi-colon with period

Line 127: delete commas after plantation and tributaries. And consider breaking up this very complex sentence.

Line 135-136: elevational

Line 175: delete the before Worldclim

Line 224: delete 'were used to' and change to 'identified'

Line 225-227: suggest using areas or regions when defining type 1 and type 2, not mixing

Line 230: remain remnants has complex meaning that isn't needed here. I suggest just 'areas retained between the current...'. Then on line 232, could say retained and new areas (although better just to say type 1 and type 2)

Line 232-234: this is confusing - was this done additionally? if not, it should be said earlier. this is the methods, and definitions have already been given, so this whole section should be streamlined to focus on what was done methodologically, not conceptually.

Line 235-236: refuge means something else. use refugia or refugium (singular). and no need to add 'areas'

Line 239: how were low and high footprint defined? and again, this and the following sentences are focusing on explanations and concepts and should be focused on details and methodology

Results: It would be good to list the mean. Maybe after 'scenarios,' state 'the average range loss across all species was X (range: x-x). Then you can talk about the 8 of the 12...

Also, with "(Fig 2)" list some stats about the footprint results

Figure 3 & 4: add "(m)" to y axis

6. PLOS authors have the option to publish the peer review history of their article (what does this mean? ). If published, this will include your full peer review and any attached files.

**Do you want your identity to be public for this peer review?** For information about this choice, including consent withdrawal, please see our Privacy Policy .

Reviewer #1: No

Reviewer #2: **Yes: ** Toni Lyn Morelli

---

## [Author Response · Author response to Decision Letter 1]

18 Feb 2025

Additional Editor Comments:

Review by the editor (Dárius Tubelis):

1) Title page. Please check again the instructions regarding the authorship and affiliations.

Affiliation: start with the Department, and then the University, to the country (see the wrong item 1).

R/Solved

2) Abstract. Line 22. Is "poorly" really correct here ? Maybe a too strong term.

About line 25 and ahead. Could you provide 1-2 sentences regarding the methods of your study ?

R/Solved

Introduction

3) You can indent the first line of the paragraphs, except the first one. (also check this for other sections of the manuscript).

R/Solved

4) The Introduction is of good lenght and quality but the number of cited studies is considerably low.

You tended to cite only one study per sentence, but you can easily increase it to 2-3 citations per brackets by making a new search in Scopus. This would increase the probability of citing the main relevant papers on this topic. Consider the PLOS ONE international audience. With this improvement, the numbers of different studies would change here and in the References Section. Please check carefully later, if you change them. You can make this improvement easily, by making little or no changes in your current text.

R/We added several references across the entire MS following this and the reviewer´s suggestions.

5) Line 85. Add "Colombia".

Methods

Study Area

6) Well done; well written.

Species Data

7) Line 142. You can add a reference that you used to classify these species as frugivores and nectarivores.

8) Line 150-154. But are the aspects of these trais in the objectives ?

Distribution Models

9) Line 185. Add the valley/province and Colombia to the Table title.

Table 1. Column "Variable". Should "tres" be "trees" ?

Results

10) Lines 262-263. "reduction of between" sounds worng. Maybe you can write "reduction of "1-75%".

11) Line 269. Fix the "and" in italics. The same for line 278.

12) Figures 2, 3 and 4. Their captions. Add the valley/province and Colombia in the end of the first sentence. Check this for all figures and tables.

13) Line 314. You have to bring the captions of figures 5 and 6 here, after this paragraph where you firstly cited them.

Discussion

14) Within brackets, you are adding a space between consecutive numbers. It should not occur. Please check throughout the manuscript. It should be, e.g., [2,3].

You have to divide the Discussion in sub-sections with subtitles. Preferably, if you follow those of the Results, and one more regarding Conservation Issues.

Keep your mind always with an international readership.

References

15) Journal titles should be abbreviated.

When citing the pages of articles, you have to use a long dash, not hyphen between numbers.

Please check again if everything is in agreement with the Instructions before sending the corrected version.

Supporting Information

16) Please make sure that the captions and titles inform that you studied birds, in the Magdalena valley/province, in Colombia.

R/Solved

Figures

17) It is difficult to read what is written there. Maybe only a problem due download. It will be checked during the next stages.

Figures 3 and 4. I did not understand why you used letters here, instead of the numbers of each group.

Figures 5 and 6. These two black circles are useless as they are. Eliminate them or add the names of cities. Explain in the captions the meaning of "F. G. richness".

Figures 7 and 8. Add Venezuela and move Colombia to the right (gray area). Put them in bold and bigger than the names of the two cities.

Figures 7 and 8. I did not understand why you have two color legends in each figure.

R/Solved. The additional box does not correspond to legend color but to each category of the refuges.

Reviewers' comments:

Comments to the Author

Reviewer #1:

18) While assessing the effects of climate change on functional groups of birds is relevant for biodiversity conservation in biodiversity hotspots, the study lacks novelty. Numerous articles have already been published on the impacts of climate change on bird distributions. Although the authors claim there are very few studies exploring the effects of climate change on functional groups of birds, this assertion is questionable; I found several articles on this topic with a quick Google search (see specific comments below).

R/Solved

19) The authors mention that they also evaluated the impacts of deforestation on species occurrences, but their methodology lacks clarity. They state they used a footprint layer for this analysis, yet they do not explain essential characteristics of this layer (e.g., whether it allows for species occurrence in non-forested areas, its resolution, etc.).

R/ Solved

20) Similarly, the authors omit crucial information regarding the methods they used to project species' potential distributions. I also believe that their models could be significantly improved (please see specific comments below).

R/ Solved

21) The discussion is weak. Although it is natural for the study to focus on the implications of the results within the study area, the authors have done so excessively. Given that the manuscript was submitted to an international journal, it should include more comparisons with studies conducted in other tropical regions. In this regard, I have suggested some relevant studies in the comments below.

R/ We added general overview and compared with broader studies.

22) The authors also fail to discuss the implications of their results for the potential movement of species under future scenarios. For instance, is the establishment of ecological corridors relevant for species movement within the landscape? Where could these potential corridors be implemented?

R/ Solved

Specific Comments:

23) Title: I suggest removing "in a degraded area of Colombia." Including regional characteristics in the title might not appeal to an international audience, although this information should be included in the abstract.

R/Although we appreciate the comment, mentioning “degraded” is key to the contents of the paper, and do not refer to a specific region of the country but to a state of the landscape, which can be of interest elsewhere. We would prefer to retain the concept.

24) Figures: The text in the figures, especially in Figures 1, 2, 3, and 4, is difficult to read. This might be due to the low-resolution images required by the submission system, but I couldn't properly review them. Additionally, is there a reason why the figure legends are spread throughout the manuscript? This arrangement makes them hard to find. Please consider adding a small inset map of Colombia showing the location of the study area. For readers unfamiliar with Colombia's geography (like myself), the location of Magdalena Province might not be immediately clear.

R/Thanks, added and a S1 Figure added.

25) Lines 34-35: I suggest excluding "for other wildlife."

26) Line 35: Consider removing "highly," as it sounds a bit vague.

27) Line 35: Replace "value" with "importance."

28) Keywords: Unless it is a journal requirement, I suggest using lowercase for all keywords and omitting the full stop.

29) Lines 59-60: "and rarely consider their effect from a functional point of view"—but see the following studies:

30) Lines 114-117: Instead of providing biome names, it might be more informative to describe the vegetation types (e.g., forests, savannas, shrublands, etc.).

31) Lines 140-142: From where did you obtain the species list of 494 species that may occur in the region? Which classification system did you follow to categorize them as frugivorous or nectarivorous? Please clarify, even if it was based on personal knowledge.

32) Line 143: Why did you choose a 30-year timeframe?

33) Lines 145-146: Does this mean you retained species records that are more than 1 km apart? If so, could this result in occurrence points being, for example, 1.5 km apart or slightly more? This could lead to high spatial correlation of your occurrence points, potentially causing model overfitting and biasing the results toward regions with more species occurrence data.

34) Lines 150-154: Please clarify why you selected those particular measures as proxies for functional traits.

35) Lines 176-178: Please explain why you chose these bioclimatic variables. How did you handle correlation among variables?

36) Line 181: Is your land cover class variable categorical? It's generally not advisable to use categorical variables in species distribution modeling (SDM) studies.

37) Line 187: Why did you choose Maxent? Why not include other algorithms and perform an ensemble approach?

38) Lines 189-190: "five regularization values, and five feature combinations"—I don't understand what this means; please clarify.

39) Line 191: I'm not sure if AUC is the best evaluation metric. Why didn't you use TSS? https://doi.org/10.1016/j.ecolind.2022.109830

R/ All solved

40) Lines 201-206: How did you perform the ensemble of your global circulation models?

R/ No assemblies of the circulation models were created, as projections for each climate variable from the circulation models available in WorldClim were used directly. Each global circulation model was evaluated separately.

41) Lines 211-212: Was the convex hull created for each species individually? If so, I agree. Please clarify.

R/ Lines 222-225 clarify that the convex hull was constructed using the combined data of all species comprising each functional group. In other words, one convex hull per group.

42) Lines 215-219: I didn't understand how you considered the "effect of landscape transformation." Please provide more information about the variable you used and your analysis.

43) Lines 245-248: The number of species in each group, as I summed from Table S1 and Figure 1, was 58 for frugivores and 53 for nectarivores. In line 140, you mention that you estimated the distribution of 494 species. Please clarify.

44) Line 264: I suggest moving Figure 2 to the Supplementary Material.

45) Lines 267-275: Since species in each group can be seen in Figure 1 and Table S1, I suggest removing species names from the text.

R/ All solved

Reviewer #2 (Toni Lyn Morelli):

46) This is a robust analysis in an important biodiversity hotspot that deserves more attention. The SDM methodology is traditional but sound. The focus on functional groups is a nice novel addition to this deep area of research/analysis.

R/ Thanks for the comment!

47) I have some medium-sized comments and then a lot of minor wording suggestions below. In general the manuscript is really well written but there are some sections that could be clearer. This could be accomplished by doing a close read and trying to get rid of extra words. I've noted some below. Here's another example: Line 316: "Analysis of projected climate change scenarios identified potential refugia for frugivorous birds. We categorized these refuges based on human footprint and protected area status" could be more directly reworded as "Extensive potential climate refugia were identified for frugivorous birds: Type 1 refugia:...

R/ Thanks for the suggestion we tried to address it across the ms.

48) 2. The commonly used terms in the (English) literature now are climate refugia and climate change refugia. Best to pick one and use throughout.

R/ Solved.

49) 3. The Methods are not detailed enough and too conceptual. I gave some examples below.

R/ Solved.

50) 4. Discussion could use some more caveats. For example, related to Line 221: how does the 'assumption was used that this human footprint index would remain constant in future scenarios.' affect the results? Also (~272-274), need to note that SSP585 is increasingly considered unlikely (this is new science) so these results are low probability, worst case scenario. On that point, the SSP 126 results should be showcased a bit more in the text to show the range of possibilities. Also, Lines 321-322 in the Results belong in the Discussion.

R/ Solved

Minor suggestions:

51) Line 21: Suggest 'drivers of the biodiversity crisis' to avoid saying loss again

52) Line 29: Use altitude or elevation consistently throughout (apparently elevation is the more appropriate term)

53) Keywords: I suggest simply 'functional group' since 'Avian functional groups' is likely too specific to be a useful keyword term

54) Line 44: Insert comma before since

55) Line 45: reword - can delete 'the population sizes of' or reword as 'there has been a 68% reduction in the populations of all (known) mammals..." nearly is not needed unless you want to say 70%

56) Line 48: wording makes it clear that the previous paragraph was supposed to be about habitat loss, but besides for the opening sentence, the paragraph just seems to be about the biodiversity crisis in general.

57) Line 57: change comma after however to semi-colon

58) Line 58: delete 'relative approaches to'

59) Line 64: change mitigate to reduce - avoid using the term mitigate in the climate change context when not referring to greenhouse gases to avoid confusion

60) Line 71: replace semi-colon with period

61) Line 127: delete commas after plantation and tributaries. And consider breaking up this very complex sentence.

62) Line 135-136: elevational

63) Line 175: delete the before Worldclim

64) Line 224: delete 'were used to' and change to 'identified'

65) Line 225-227: suggest using areas or regions when defining type 1 and type 2, not mixing

66) Line 230: remain remnants has complex meaning that isn't needed here. I suggest just 'areas retained between the current...'. Then on line 232, could say retained and new areas (although better just to say type 1 and type 2)

67) Line 232-234: this is confusing - was this done additionally? if not, it should be said earlier. this is the methods, and definitions have already been given, so this whole section should be streamlined to focus on what was done methodologically, not conceptually.

68) Line 235-236: refuge means something else. use refugia or refugium (singular). and no need to add 'areas'

69) Line 239: how were low and high footprint defined? and again, this and the following sentences are focusing on explanations and concepts and should be focused on details and methodology

70) Results: It would be good to list the mean. Maybe after 'scenarios,' state 'the average range loss across all species was X (range: x-x). Then you can talk about the 8 of the 12...

71) Also, with "(Fig 2)" list some stats about the footprint results

Figure 3 & 4: add "(m)" to y axis

R/ All solved.

---

## [Editor Report · Decision Letter 1]

12 Mar 2025

Functional safeguards for conservation: identifying climate change refugia for frugivorous and nectarivorous birds in a degraded area of Colombia

PONE-D-24-31667R1

Dear Dr. José González-Maya,

We’re pleased to inform you that your manuscript has been judged scientifically suitable for publication and will be formally accepted for publication once it meets all outstanding technical requirements.

Kind regards,

Dárius Pukenis Tubelis, Ph.D.

Academic Editor

PLOS ONE

Additional Editor Comments:

Dear Dr José F. González-Maya,

I received the corrected version of your submission PONE-D-24-31667 by Dr Fausto Sáenz-Jiménez et al about refugia for Colombian birds.

Sorry for taking 11 days to send my decision, but my University had no internet for 8 days this month.

Thus, I could not handle the corrected manuscript immediately.

I carefully read the corrected version (R1) and noted that you followed most of the numerous suggestions, comments and corrections

provided by both reviewers and me. Also, I appreciated most of your responses to all points presented by us.

With this, I now consider that your study reached the expected standard for publication in PLOS ONE.

Your article is in agreement with the PLOS ONE Publication Criteria.

Thus, my decision is to Accept your submission.

During my last reading of your corrected version, I found a range of minor mistakes that have to be fixed prior to publication.

You can fix then along the next days and final stages of the evaluation process.

You will find my final corrections below.

Thank you for considering PLOS ONE as home of your research.

Dárius P. Tubelis

PLOS ONE Editor

UFERSA

Mossoró-RN Brazil

**Last corrections to be done by authors prior to publication:**

First page, authors. Replace "&" by a comma prior to the last author.

Abstract

Line 22. "remain understood" has to be fixed. Maybe you can use "have to be better understood". Or other option to replace the former "poorly".

Introduction

Line 61. This "; however," is a bit strange. Can you delete it ? Please check.

Line 76. Add a space before "Furthermore".

Line 90. I think it should be "Colombian". Add "n".

Methods

Line 166. Delete "." after "interactions".

Table 1. Line 200+, fix "tres" for Evergreen broadleaf, second column.

Results

Lines 273-278. Please fix the formatting of this paragraph. It is to the left.

Line 280. Replace "Figure 1." by "Fig 1." Please do this for all figure captions.

Line 292. Delete the dot after "Figs".

Line 303. The same.

Line 307. "and" should not be in italics.

Line 313. Delete the dot after "Figs".

Lines 315 and 318. Use "Fig" instead of "Figure". The dot comes only after the number.

Lines 325-327. Delete the dot after "Fig".

Lines 329 and 332. Use "Fig" instead of "Figure".

Line 240. Replace "Fig." by "Figs".

Lines 343 and 347. Use "Fig" instead of "Figure".

Line 356. Use "Fig 8".

Line 362. Use "Fig 9".

Lines 366 and 372. Use "Fig" instead of "Figure".

Discussion

Line 395. I think there is no need to start a new (short) paragraph. Consider merging that of "Beyond" with the previous paragraph.

Lines 404 to 407. Other too short paragraph. Merge it with the previous or next one.

Line 426. Maybe you can use "Cracids" as you use "are" ahead.

Lines 431 to 434. Other short paragraph. Consider merging with the previous one.

Line 455. Delete the dot after "m". And add a space.

Line 456. Delete the dot after "Fig". Use "Figs 4 and 5".

Line 502. Use "(Figs 8 and 9)".

Line 516. You can add another study besides 123 or change to "a study in Brazil [123] has...." . Similar to lines 530-531.

Line 526. Use a long dash between numbers.

References

I noted that the formatting was substantially improved. I found not mistakes, but please make a final checking before sending the final version.

Supporting information

There is some **GREAT** problem here.

You do not provide the titles for S1 Table, S2 Table and S3 Table. They were cited on the text but their titles should be here in this list. Also, their files are not in the ZIP FILE ARCHIVE.

S4 Table. It was not cited on the text and its title is not in this list of titles.

S5 Table, S6 Table, S7 Table and S8 Table. Their titles are here but they were not cited on the text.

Please fix these major problems carefully.

FIGURES

Where is Fig 1 ? Your figures start with Figure 2.

Please check the numbers of all figures again. Mistakes should not occur here. I´m confused with it.

Figures 8 and 9. The word "Colombia" is partially covered. You have to fix it maybe by reducing the size of the box with elevation data.

---

## [Editor Report · Acceptance letter]

PONE-D-24-31667R1

PLOS ONE

Dear Dr. González-Maya,

I'm pleased to inform you that your manuscript has been deemed suitable for publication in PLOS ONE. Congratulations! Your manuscript is now being handed over to our production team.

Kind regards,

on behalf of

Dr. Dárius Pukenis Tubelis

Academic Editor

PLOS ONE